# Daytime-Restricted Feeding Ameliorates Oxidative Stress by Increasing NRF2 Transcriptional Factor in the Rat Hippocampus in the Pilocarpine-Induced Acute Seizure Model

**DOI:** 10.3390/brainsci13101442

**Published:** 2023-10-10

**Authors:** Octavio Fabián Mercado-Gómez, Virginia Selene Arriaga-Ávila, Angélica Vega-García, Sandra Orozco-Suarez, Vadim Pérez-Koldenkova, Juan José Camarillo-Sánchez, Marcelino Álvarez-Herrera, Rosalinda Guevara-Guzmán

**Affiliations:** 1Department of Physiology, School of Medicine, National Autonomous University of Mexico, Mexico City 04510, Mexico; omercado@comunidad.unam.mx (O.F.M.-G.); vsarriagaa@unam.mx (V.S.A.-Á.); ange_li_k@hotmail.com (A.V.-G.); juan.jose.cs81@gmail.com (J.J.C.-S.); resorte2@hotmail.com (M.Á.-H.); 2Medical Research Unit in Neurological Diseases, National Medical Center XXI, Mexico City 06720, Mexico; sorozco5@hotmail.com; 3National Advanced Microscopy Laboratory, National Medical Center XXI, Mexico City 06720, Mexico

**Keywords:** daytime-restricted feeding, oxidative stress, status epilepticus, Nrf2, SOD2, hippocampus

## Abstract

Seizure-mediated oxidative stress is a crucial mechanism in the pathophysiology of epilepsy. This study evaluated the antioxidant effects of daytime-restricted feeding (DRF) and the role of the Nrf2 signaling pathway in a lithium-pilocarpine model seizure model that induces status epilepticus (SE). We performed a lipoperoxidation assay and dihydroethidium fluorescence to measure oxidative stress markers in the hippocampus (malondialdehyde and reactive oxygen species). The protein content of Nrf2 and its downstream protein SOD2 was evaluated using Western blotting. The cellular distribution of the Nrf2 and SOD2 proteins in the pyramidal cell layer of both the CA1 and CA3 hippocampal subfields and astrocytes (GFAP marker) were quantified using immunofluorescence and immunohistochemistry, respectively. Our results indicate that DRF reduced the malondialdehyde levels and the production of reactive oxygen species. Furthermore, a significant increase in Nrf2 and SOD2 protein content was observed in animals subjected to restrictive diet. In addition, DRF increased the relative intensity of the Nrf2 fluorescence in the perinuclear and nuclear compartments of pyramidal neurons in the CA1 subfield. Nrf2 immunoreactivity and the astrocyte marker GFAP also increased their colocalization under DRF conditions. Additionally, SOD2 immunoreactivity was increased in CA1 pyramidal neurons but not in the CA3 region. Our findings suggest that DRF partially prevents oxidative stress by increasing the Nrf2 transcriptional factor and the SOD2 enzyme during the development of SE.

## 1. Introduction

Epilepsy is the third most common chronic brain disorder, affecting nearly 70 million people. It is characterized by recurrent spontaneous seizures due to the hyperexcitability of brain neurons [1]. *Status epilepticus* (SE) results from the failure of the mechanisms responsible for seizure termination or initiation, leading to prolonged seizures within a period of 5 min that can occur or not in epileptic patients [2]. SE must be treated urgently; otherwise, it could trigger serious consequences such as neuronal death, neuronal injury, and alteration of neuronal networks [2,3].

Oxidative stress has recently been recognized as playing a crucial role in the pathophysiology of SE and epilepsy [4,5,6]. Oxidative stress results in functional cellular damage and may cause subsequent cell death via the oxidation of biomolecules such as proteins, nucleotides, and lipids [4]. Prolonged seizures generate reactive oxygen species (ROS); this process is carried out by nicotinamide adenine dinucleotide phosphate (NADPH) oxidase and NMDA receptor activation [7]. Furthermore, the seizure-induced inflammatory response can activate inducible nitric oxide synthase to produce nitric oxide, which may react with the superoxide radical to form reactive nitrogen species (RNS) like the peroxynitrite radical. These RNS contribute to the severity of oxidative stress in the pharmacological model of SE [7,8].

Nuclear factor erythroid 2-related factor 2 (Nrf2) is a transcriptional factor related to the natural cellular defense system. Nrf2 induces the gene expression of numerous ROS-eliminating enzymes [9]. Under physiological conditions, Nrf2 is targeted for proteasomal degradation in the cytosolic compartment; however, upon oxidative or electrophilic stress, it translocates to the nucleus, binds to the antioxidant-responsive element (ARE) sequence, and promotes the transcription of antioxidant enzymes, including NAD(P)H quinone oxidoreductase 1 (NQO1), heme oxygenase 1 (HO1), and superoxide dismutase 2 (SOD2) [10]. Several recent reports have demonstrated that the Nrf2-ARE signaling pathway could represent an important target for protecting neurons after ischemic damage [11]. However, the possible protecting role of Nrf2 has not been fully studied in epilepsy.

Many metabolic-based therapies have been tested in patients with epilepsy in clinical trials and in pharmacological models of temporal lobe epilepsy. These therapies include ketogenic diets, calorie restriction, and intermittent fasting [12]. In this regard, our group has shown that daytime-restricted feeding (DRF), an intermittent fasting schedule, has an anticonvulsant effect attributed to metabolic activation (increased AMP activating-protein kinase), epigenetic mechanisms (increased histone 3 acetylation) [13], and anti-inflammatory and neuroprotective effects (preventing the activation of microglia and astrocytes) [14]. Hence, this study aimed to determine whether the DRF schedule could have an antioxidant effect after a pilocarpine SE-induced seizure and whether such an effect is due to the increase in antioxidant-related enzymes.

## 2. Materials and Methods

### 2.1. Experimental Animals

Eighty adult male Wistar rats (n = 5 per group) weighing approximately 250–280 g were used. The rats were maintained at a 12 h light/12 h dark cycle under constant temperature and humidity conditions (25 °C and, 50–70%, respectively). The animals were fed with the standard diet of Lab Diet Rodent Laboratory Diet 5001 pellets (PMI Nutrition International, Inc., Brentwood, MO, USA) and had free access to water.

### 2.2. Daytime-Restricted Feeding and the Pilocarpine-Induced Seizure Model

The experimental animals were randomly assigned to one of the following four groups: (1) a control group with ad libitum access to food (AL), (2) a pilocarpine-induced SE group with ad libitum access to food (ALSE), (3) a group under daytime-restricted feeding (DRF), and (4) a pilocarpine-injected DRF group (DRFSE). As previously described [13,14], DRF consisted of giving the rats access to food for only two hours daily for 20 days (from 12 to 2 pm) with free access to water and, after this period, we proceeded to perform the acute seizure model at day 21 (Figure 1).

We chose the lithium-pilocarpine model because it is one of the most widely used models to induce SE [13,14]. The animals were first injected with lithium-chloride (127 mg/kg, i.p.) on day 20; 18 h later, they received a scopolamine methyl nitrate injection (1 mg/kg, s.c.) to minimize the peripheral cholinergic effects of pilocarpine. Thirty minutes later, pilocarpine chloride was administered (30 mg/kg, s.c.) to induce SE. Ninety minutes later, the seizures were stopped with a diazepam injection (5 mg/kg i.m.).

The behavioral grading of the seizures was performed via video monitoring for 2 h. The scoring was based on Racine’s scale [15] with the following stages: (0) no abnormality; (1) akinesia and facial movements; (2) head nodding; (3) forelimbs clonus; (4) rearing; and (5) rearing and falling. Only the animals that reached stages four to five were used; the rest were discarded. The AL control and DRF animals only received a saline injection instead of a pilocarpine injection. Importantly, the food was removed 12 h before the usual start of food restriction to ensure that all animals were in the same metabolic condition. Therefore, SE was induced after 6 h of fasting in the AL-pilocarpine rats and after approximately 22 h of fasting in the DRF-pilocarpine rats. Moreover, both AL- and DRF-pilocarpine animals received a saline solution injection to avoid dehydration. Twenty-four hours after the pilocarpine injection, the animals were euthanized with an overdose of sodium pentobarbital (26 mg/kg) to perform biochemical analyses (Figure 1). All the experiments from the present study were approved by the Ethical Committee of the Faculty of Medicine at UNAM following all the statements to minimize animal suffering.

### 2.3. Determination of Lipid Peroxidation

The rat hippocampi were dissected as quickly as possible and were immediately washed with phosphate-buffered saline solution (PBS, pH 7.4) and transferred into clean plastic tubes. To measure lipid peroxidation products such as malondialdehyde (MDA) from hippocampal tissue, we used the ALDetect lipid peroxidation assay kit (Enzo Life Science, Farmingdale, NY, USA, BML-AK170) following the manufacturer’s instructions. The results were expressed as nmol of MDA per mg of protein.

### 2.4. Determination of Reactive Oxygen Species in Hippocampi

The rats’ brains were quickly dissected after euthanasia and frozen with butanol at −70 °C. Coronal slices (20 μm) were obtained in a cryostat and placed in slides with oly-L-lysine. The slides were incubated with a solution of dihydroethidium (DHE, 10 μM) at room temperature for 30 min and protected from light. Once the incubation time had elapsed, the sections were mounted and observed under an epifluorescence microscope. Although DHE has been described to measure superoxide anion, it also can detect other reactive oxygen species such as hydrogen peroxide and hydroxyl radicals.

### 2.5. Western Blotting

Hippocampi homogenates were transformed into cytoplasmic extracts using a commercial buffer lysis (Thermo-Fisher Scientific, Waltham, MA, USA). Homogenate samples were collected and stored at −70 °C for later analysis. Protein extracts were quantified using a BCA assay kit (Pierce, Appleton, WI, USA). For this, 60 µg of protein was loaded on 10 or 15% SDS-PAGE gels. The proteins were transferred to a nitrocellulose membrane in a wet tank transfer system (Mini Trans-Blot Central Core, Bio-Rad Laboratories, Hercules, CA, USA). Then, the membranes were rinsed with Tris-buffered saline (TBS) and blocked with a solution containing 5% non-fat dry milk in TBS-Tween 20 0.1% (TBST) overnight at 4 °C. The blots were probed with anti-rabbit polyclonal Nrf2 (1:1000, Santa Cruz Biotechnology, Dallas, TX, USA, SC-722), and anti-rabbit polyclonal manganese superoxide dismutase (SOD2) (1:1000, Boster Biological Technology, Pleasanton, CA, USA, A00349) in TBST at 4 °C for 48 h. After three rinses with TBST for five minutes each, the membranes were incubated with anti-rabbit (1:10,000 Cell Signaling Technology, Beverly, MA, USA) or anti-mouse IgG secondary antibodies (1:5000, Santa Cruz Biotechnology, SC-516102) for 90 min at room temperature, followed by three rinses with TBST for five minutes each. As loading control, we used mouse monoclonal antibody against beta-actin (1:1000, Genetex, Irvine, CA, USA, GTX629630). The membranes were revealed using the chemiluminescence reagent Luminata Crescendo Western HRP, Millipore. Images were obtained with the Fusion FX photo documentation device (Vilber, Lemont, IL, USA). Image analysis was performed with the Fiji image processing software version 1.54 developed at the National Institutes of Health and available online (https://imagej.net/software/fiji/, accessed on 20 November 2022). The data obtained from the density analysis of each protein under study were normalized with the data of its corresponding protein.

### 2.6. Immunofluorescence and Immunohistochemistry

The animals were euthanized as previously mentioned and perfused transcardially with 300 mL of ice-cold PBS followed by 250 mL of 4% paraformaldehyde in phosphate buffer (pH 7.4) as a fixative solution. Their brains were removed and immersed in the fixative solution overnight. Then, the brains were dehydrated using alcohol solutions and xylene and embedded in paraffin wax. Coronal sections of 5 μm of thickness were cut in a microtome and placed in poly-L-lysine coated slides. The paraffin was removed, and the sections were pretreated with a heat retrieval Diva Decloacker solution and placed in an electric pressure cooker (decloaking chamber, Biocare Medical, Pacheco, CA, USA) for 15 min. The coronal sections were rinsed with distilled water and PBS. Moreover, to avoid autofluorescence of the sections, we submerged the slides into a Coplin glass filled with a saturated solution of Sudan black B (0.25%) in 70% isopropyl alcohol. The hippocampal sections were rinsed with PBS for 5 min, permeabilized with PBS Triton-100× (0.3%) for 20 min, and blocked with 1% normal goat serum (Vector Laboratories, Inc., Newark, CA, USA) in the PBS solution for 30 min. Then, rabbit polyclonal Nrf2 (1:100, Santa Cruz Biotechnology, SC-722) and anti-mouse antibody against the glial fibrillary acidic protein (GFAP, 1:200, Biocare Medical, CM065) were incubated in blocking solution at 4 °C for 48 h. After three rinses in PBS, the sections were incubated with anti-mouse Alexa Fluor 488 and Alexa Fluor 594 conjugated donkey anti-rabbit IgG antibody (1:300, Thermo Fisher Scientific) in the blocking solution for 2 h at room temperature. Then, the slides were rinsed three times with PBS (5 min each). After washing, the nuclei were labeled with Hoescht (1:5000, Roche Lab, Indianapolis, IN, USA) in PBS (for 1 min. The tissue was washed again with PBS for 3 min and covered. For immunohistochemistry, the coronal sections were permeabilized with PBS containing 0.3% Triton X-100 and 0.3% H_2_O_2_ solution for 30 min and left with the blocking solution (Background Sniper, Biocare Medical) for 1 h to reduce the background staining. Then, the brain sections were incubated with rabbit polyclonal anti-SOD2 (1:300, Boster Biological Technology, A00349) overnight at 4 °C and rinsed three times for 5 min with PBS. Immediately after, the sections were incubated with Trekkie Universal Link for 1 h at room temperature rinsed with PBS again and then with TrekAvidin-HRP (Starr Trek Universal HRP Detection, Biocare Medical) for 1 h, and rinsed one more time with PBS. Afterwards, coronal sections were revealed by using 3, 3-diaminobenzidine (Betazoid DAB Chromagen Kit, Biocare Medical) dehydrated, mounted, and observed under a Brightfield microscope (Leica Microsystem, Wetzlar, Germany). The negative controls underwent the same procedure but without the primary antibodies in both techniques.

### 2.7. Confocal Microscopy Analysis

The brain sections were evaluated with a Nikon Ti Eclipse inverted confocal microscope equipped with an A1 imaging system, both controlled from the proprietary NIS Elements v.4.50 software. Imaging was performed using a 20X objective (dry, NA 0.8). The dye was excited in a sequential mode using the integrated laser lines: 403 nm (Hoechst), 488 nm (Alexa 488), and 620 nm (Alexa 594). The corresponding fluorescence was read in the following ranges: 425–475 nm (Hoechst 33342), 500–550 nm (Alexa fluor 488), and 570–620 nm (Alexa fluor 594) using the manufacturer-provided filter sets. Images were acquired and analyzed using NIS Elements v.4.50. The intensity of the red channel pixels (Alexa 594) per area was quantified with FIJI software. The intensity per area was calibrated using a spatial scale derived from the maximum and minimum intensity values contained in the bitmap of each image (image provided by the software). This calibration allowed us to establish the basal intensity values of each image. The density was calculated in a similar manner to the intensity per area; it was determined relative to the control groups and expressed in pixels/mm^2^. The count area was adjusted to 1 mm^2^.

### 2.8. Statistical Analysis

All data are presented as the mean ± standard deviation of the mean (S.D.) and examined with the appropriate normality test. A one-way ANOVA with Tukey’s multiple comparison test or the Kruskal–Wallis test followed by Dunn’s post hoc test were used to measure the optical band’s density from the Western blot assay and the relative intensity from DHE fluorescence and Nfr2 nuclear factor immunofluorescence. All statistical tests were performed using GraphPad Prism statistics software version 7 (GraphPad Software, San Diego, CA, USA), and *p* < 0.05 was considered statistically significant.

## 3. Results

### 3.1. Dietary Restriction Reduces the Levels of Malondialdehyde in Seizure-Induced Animals

Since SE has been associated with an increase in lipid peroxidation, we evaluated the levels of malondialdehyde (MDA) in the hippocampal tissue of the experimental groups. There was a significant increase in the MDA levels in the AL-injected pilocarpine group compared with the AL control group (*p* < 0.001, Figure 2). Remarkably, we found a significant decrease in MDA levels between the AL-injected pilocarpine and DRF-injected pilocarpine groups (*p* < 0.01, Figure 2). Moreover, the MDA levels were slightly higher in the DRF-schedule group than in the AL control group, but this was not significant.

### 3.2. Daytime-Restricted Feeding Reduces the Levels of Reactive Oxygen Species in Seizure-Induced Animals in CA1 and CA3 Hippocampal Regions

Using DHE fluorescence, we found that the reactive oxygen species (superoxide anion, hydrogen peroxide, and hydroxyl radical) levels were significantly increased in the CA1 subfield of the hippocampus in the ALSE group compared to the AL control group (*p* < 0.001, Figure 3A,B). Moreover, the DRFSE group significantly reduced the relative intensity of DHE fluorescence to the levels of the ALSE group (*p* < 0.01, Figure 3A,B). Regarding the CA3 subfield, we also observed a significant increase in the relative intensity of DHE in the ALSE group compared to their respective control (*p* < 0.05, Figure 3C,D). Furthermore, the restrictive diet did not statistically reduce DHE relative intensity in the DRFSE group compared to the ALSE group (Figure 3C,D).

### 3.3. Daytime-Restricted Feeding Modulates the Content of the Nrf2 Transcriptional Factor in Hippocampal Homogenates and Increases the Nrf2 Immunostaining in CA1 and CA3 Pyramidal Neurons in the SE Model

Since DRF induced a significant decrease in lipoperoxidation and a decrease in the relative density of DHE fluorescence, we evaluated whether the Nrf2 nuclear factor could be involved in the defense against cellular oxidative stress. The ALSE group showed a significant increase in cytoplasmic Nrf2 protein content compared to that in the AL and DRF groups (*p* < 0.0001, Figure 4A,B). Moreover, the restrictive diet increased the Nrf2 protein content after pilocarpine-induced seizures compared to the ALSE group (*p* < 0.0001, Figure 4A,B).

To further understand the protective role of the Nrf2 protein against oxidative stress, we studied its cellular distribution in the CA1 and CA3 hippocampal subfields since these brain regions become widely damaged in temporal lobe epilepsy [16]. As shown in Figure 4C, the relative intensity of Nrf2 in the CA1 hippocampal region was higher in animals subjected to SE than in the AL control or DRF groups. In particular, there was a high intensity in the perinuclear compartment (arrows) and a low intensity inside the nucleus (*p* < 0.0001, Figure 4C,D, arrowheads). Interestingly, the DRFSE group had higher Nrf2 relative intensity than the ALSE group (*p* < 0.001, Figure 4C,D). Furthermore, Nrf2 immunostaining in the DRFSE group was more intense in the perinuclear compartment (arrows) and showed medium intensity in the nucleus of the pyramidal cell layer (arrowheads).

Kim and Kang [17] recently showed that an Nrf2 activator prevents the loss of Nrf2 nuclear factor in astrocytes in rats with chronic epilepsy. Thus, we detected glial fibrillary acid protein (GFAP), a well-known astrocyte marker through immunofluorescence, to assess whether the restrictive diet might have a similar action. GFAP immunoreactivity was more abundant in the *stratum radiatum* and cells surrounding the pyramidal cell layer of CA1 in the DRFSE group than in the ALSE group (Figure 4B). Furthermore, we found a major overlapping of Nrf2 and GFAP signals in astrocyte processes in the DRFSE group versus the ALSE group (Figure 4B).

Similarly, in the hippocampal CA3 subfield, Nrf2 intensity was higher in the rats injected with pilocarpine compared to their respective controls (*p* < 0.0001, Figure 4E,F). Nrf2 immunostaining was localized in the perinuclear compartment, but unlike what was observed in CA1, the Nrf2 signal in this region was punctate and discontinuous. We also observed the Nrf2 signal in some nuclei of pyramidal neurons (Figure 4E). In the case of the DRFSE group, we did not observe significant changes in Nrf2 fluorescence intensity compared with the ALSE group, although there was a slight increase in the Nrf2 signal (Figure 4E,F). Similar results were obtained for GFAP immunoreactivity, in which the overlap of the Nrf2 and GFAP signals was more evident in the DRFSE group than in the ALSE group; however, such an observation was merely qualitative (Figure 4E).

### 3.4. Daytime-Restricted Feeding Increases the Content of Superoxide Dismutase 2 in Hippocampal Homogenates and the Immunostaining in CA1 and CA3 Pyramidal Neurons after the Acute Seizure Model

Due to the fact that superoxide dismutase 2 (SOD2) is a downstream protein of the Nrf2 signaling pathway [10], we evaluated the mitochondrial SOD2 levels in the hippocampal homogenates through Western blotting. In this regard, we observed that the ALSE group showed a significant increase in SOD2 protein content compared to the AL and DRF groups (*p* < 0.0001, Figure 5A,B). Notably, daytime-restricted feeding increased the SOD2 protein content after pilocarpine-induced seizures compared to the ad libitum pilocarpine-induced seizure (*p* < 0.0001, Figure 5A,B). Then, we evaluated the relative intensity of the SOD2 immunoreactivity of the pyramidal cell layer of the CA1 and CA3 subfields. Unlike the previous results in the SOD2 protein content in the hippocampal homogenates, we observed a significant decrease in the relative density of SOD2 immunostaining in pyramidal neurons of CA1 in the ALSE group compared to the AL group (*p* < 0.001, Figure 5C,D, arrows). Interestingly, the restrictive diet was able to significantly increase the immunoreactivity of SOD2 in the pyramidal cell layer of CA1 compared to the ad libitum pilocarpine-induced seizure (*p* < 0.001, Figure 5C,D, arrows). Similar results were observed in the CA3 region. There was a significant decrease in the relative density of SOD2 immunostaining in the pyramidal neurons of CA3 in the ALSE group compared to the AL- or DRF-alone groups (*p* < 0.001 and *p* < 0.01, respectively, Figure 5E,F, arrows). Notably, there was more of a tendency to increase the immunoreactivity of SOD2 in the DRFSE group than in the ALSE group; however, there were no statistically significant results (Figure 5E,F, arrows).

## 4. Discussion

Epilepsy is a global public health concern, and the development of new, effective pharmacological therapies has been limited. Therefore, addressing treatment efficacy is essential, especially for patients with drug-resistant epilepsy. Although many anticonvulsant drugs are available for epilepsy treatment, most target neurotransmitter systems or ion channels [5].

Oxidative stress in epilepsy results in cellular damage and the disruption of cellular function. Furthermore, it may cause cell death because neurons are particularly vulnerable to oxidant damage due to the high oxygen demand, poor repair capacity, and the presence of polyunsaturated fatty acids [18,19].

Several reports have shown that repeated seizures (SE) may induce peroxidation products due to the exacerbated production of reactive oxygen species. Likewise, numerous animal studies have demonstrated that antioxidants such as coenzyme Q10, vitamin C, N-acetyl-cysteine, and flavonoids reduce lipoperoxidation and restore the activities of different antioxidant enzymes, including superoxide dismutase, catalase, and glutathione [6].

In this study, we report that DRF ameliorates the oxidative stress induced by pilocarpine injection and that this antioxidant effect could be mediated by an increase in nuclear factor Nrf2 and its downstream protein SOD2. Our results indicate that daytime-restricted feeding can significantly reduce malondialdehyde levels in pilocarpine-injected rats (Figure 2). Furthermore, daytime-restricted feeding tends to decrease the production of superoxide radicals measured indirectly with DHE fluorescence (Figure 3). To our knowledge, this is the first report describing a potential antioxidant role for an intermittent fasting schedule such as daytime-restricted feeding in a pharmacological model of SE. However, another dietary intervention, a high-fat, low-carbohydrate diet (ketonic diet), has already been shown to display antioxidant potential in an epilepsy model [20,21].

Therefore, we focused on one of the main redox-sensitive transcription factors inducing antioxidant and detoxifying enzymes to protect cells against oxidative stress: nuclear factor erythroid 2-related factor 2 (Nrf2) [9]. Nrf2 mRNA levels are significantly upregulated in human epileptic hippocampal tissue and in the hippocampus of mice 72 h after pilocarpine injection, perhaps as an attempt to minimize the seizure-induced rise of free radicals [22]. In this regard, our results show that the seizures per se significantly increase the protein content of Nrf2 in hippocampal homogenates (Figure 4A,B). Moreover, Nrf2 immunostaining was also increased in the CA1 and CA3 hippocampal subfields (Figure 4C,E). These results are consistent with previous reports in which rats with electrically or pharmacologically induced epilepsy showed a substantial increase in Nrf2 mRNA levels and Nrf2 immunoreactivity [22,23,24]. Interestingly, recent data have shown that the activation of Nrf2 by different compounds can suppress mitochondrial oxidative stress, which mitigates seizure-induced damage [24,25,26,27]. In this regard, we hypothesized that daytime-restricted feeding could activate the Nrf2 nuclear factor. Our results indicate that daytime-restricted feeding induced a significant increase in Nrf2 protein content in the hippocampal homogenates in the pilocarpine-induced seizure group (Figure 4A). In agreement with this, the relative intensity of Nrf2 immunoreactivity in CA1 and CA3 pyramidal cells was mainly localized in the perinuclear compartment (Figure 4C,E, respectively). Importantly, the Nrf2 nuclear factor must translocate into the nucleus to bind to the antioxidant-responsive element (ARE) sequence in order to promote the transcription of downstream detoxifying enzymes [10]. In this regard, we observed a small part of the Nrf2 nuclear factor protein translocating into the nucleus in both the ALSE and the DRFSE group (Figure 4C), which correlated with Nrf2 immunostaining in some CA1 and CA3 pyramidal cell nuclei (Figure 4C,E, respectively). These results agree with previous work where kainic acid- or pentylenetetrazole-induced seizures activate an antioxidant enzyme regulated by Nrf2 [28].

Astrocytes have been widely recognized as the active partners of neurons because they modulate neuronal activity throughout the uptake and release of neurotransmitters [29]. Astrocytes also have an important role in epileptogenesis [30]. Furthermore, it is well documented that after pilocarpine-induced status epilepticus, a population of astrocytes die while others are activated, promoting astrogliosis [31]. Thus, we hypothesize that astrocytes from animals subjected to DRF could contribute to the high Nrf2 expression to improve the oxidative stress produced by repeated seizures. Recently, Kim and colleagues showed that an analog of oleanolic acid induced Nfr2 expression in astrocytes in the CA1 region and prevented astrogliosis after SE induction. Accordingly, the increased colocalization of Nrf2 and the astrocyte marker in the CA1 and CA3 subfields (Figure 4C,E) suggests that daytime-restricted feeding could have a similar effect [31].

Recent evidence has shown the relationship between oxidative stress and mitochondrial dysfunction in epilepsy. As is known, mitochondria have several key cellular functions such as the generation of ATP, calcium homeostasis, neurotransmitter biosynthesis, and the control of cell death, and they are the primary site of reactive oxygen species (ROS) [32]. Experimental models of temporal lobe epilepsy have shown an increase in ROS levels [33,34]. Mitochondrial superoxide dismutase 2 (SOD2) is a major component of the antioxidative machinery that handles ROS in the mitochondrial matrix because it determines how much superoxide radical anion (O_2_^•−^) is converted to hydrogen peroxide (H_2_O_2_) [35]. In this regard, Liang and colleagues showed that postnatal mutant mice lacking SOD2 exhibited frequent spontaneous motor seizures, providing evidence that oxidative stress-induced mitochondrial dysfunction may contribute to epileptic seizures [36]. Furthermore, it has been recently shown that specific neuronal deletion nSOD2 knockout mice develop epilepsy together with a selective loss of neurons [37]. According to our results, we observed that the SOD2 protein content increases in the hippocampal homogenates after pilocarpine-induced seizures (Figure 5A). Interestingly, DRF was able to further increase the protein content of SOD2, perhaps as an attempt to minimize the rise in ROS levels (Figure 5A). Unexpectedly, when we performed the immunohistochemistry technique to observe the cellular distribution of SOD2 protein in the pyramidal cell layer of the CA1 and CA3 subfields, we found that pilocarpine-induced seizures significantly reduced the immunoreactivity of SOD2 in both hippocampal regions (Figure 5C,E, respectively). These results could correlate with the increased levels of superoxide ion and hydrogen peroxide measured indirectly with the fluorescence of dihydroethidium (DHE) (Figure 3). Notably, DRF was able to recover the immunoreactivity of SOD2 in the pyramidal cell layer of the CA1 region (Figure 5C,D) and correlate with the reduction in the relative intensity of DHE in the same region (Figure 3A,B). Similar results were observed in the pyramidal cell layer of the CA3 region, where a pilocarpine-induced seizure decreased SOD2 immunoreactivity in neurons; however, DRF could not recover the basal levels of SOD2 after seizure induction (Figure 5E,F). These results show the crucial role of mitochondrial SOD in controlling the conversion of superoxide ion to hydrogen peroxide produced by seizures, and, most importantly, they show that DRF could downregulate the ROS levels by increasing antioxidant enzymes such as SOD2.

## 5. Conclusions

Our results suggest that daytime-restricted feeding reduces oxidative stress in the acute seizure model. This antioxidant effect could be partially mediated by the activation of the Nrf2 transcriptional factor, and the upregulation of enzymes involved in antioxidant cellular defense, such as SOD2 protein. Since ketone bodies like beta-hydroxybutyrate are produced in daytime-restricted feeding or the ketogenic diet [12,13], we cannot exclude the participation of other oxidative-stress resistance factors such as FOXO3A and metallothionein 2 A [38]. However, our data strongly support the idea that the Nrf2 protein could play a vital role in regulating oxidative stress through antioxidant enzymes such as SOD2.

In addition, we suggest the possible participation of astrocytes in modulating the expression of the Nrf2 nuclear factor to counteract the damage caused by seizure-induced oxidative stress. Our results support a possible use of DRF as an adjuvant treatment for drug-resistant epileptic patients that could improve the neurodegeneration observed in epilepsy.

## Figures and Tables

**Figure 1 brainsci-13-01442-f001:**
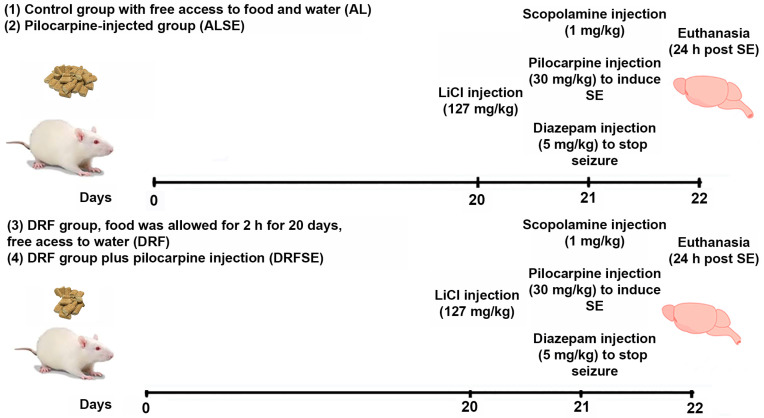
Schematic representation of the experimental procedure of the day-time restrictive feeding schedule and status epilepticus induction via pilocarpine injection.

**Figure 2 brainsci-13-01442-f002:**
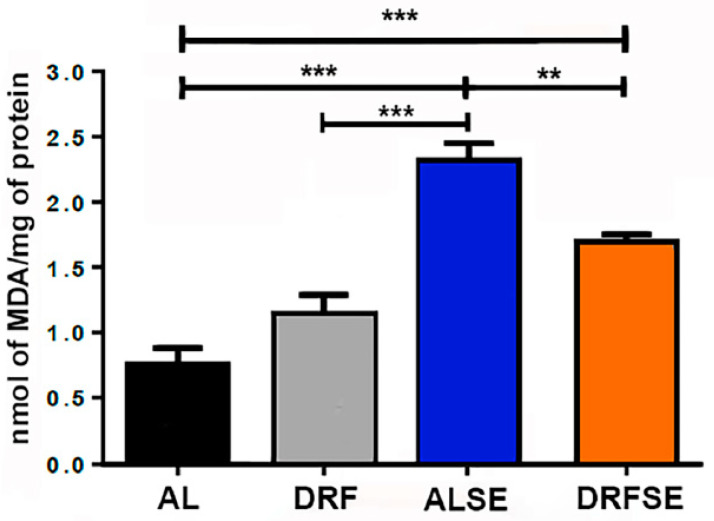
Effect of daytime-restricted feeding on lipid peroxidation after SE induction. Malondialdehyde was significantly increased in the ALSE group compared to the AL control group. The DRF group had slightly more MDA than the AL control. The DRFSE group showed lower MDA levels than the ALSE group. These results suggest an antioxidant effect of DRF. All data are presented as mean ± S.D. (n = 5 rats per group). Statistical analysis: One-way ANOVA followed by Tukey’s multiple comparison test, ** *p* < 0.01, *** *p* < 0.001.

**Figure 3 brainsci-13-01442-f003:**
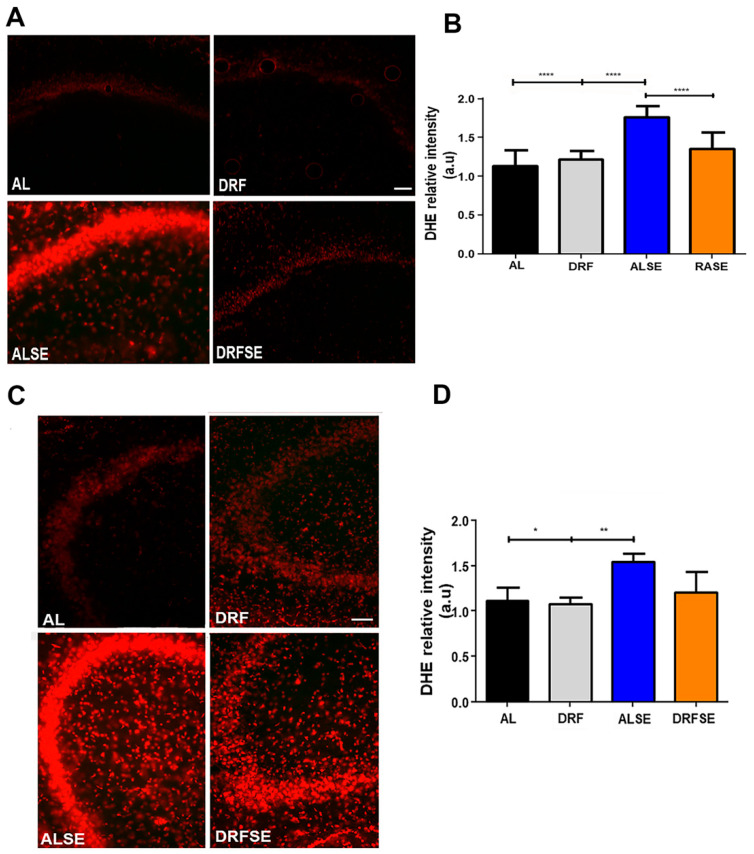
Effect of daytime-restricted feeding on superoxide radical DHE fluorescence in the CA1 and CA3 subfields of the dorsal hippocampus after SE. (**A**) Representative images of DHE-stained cells (red) in the experimental groups in the CA1 subfield. (**B**) Quantification of relative DHE intensity in the CA1 subfield. (**C**) Representative images of DHE-stained cells (red) in experimental groups in the CA3 subfield. (**D**) Quantification of relative DHE intensity in the CA3 subfield. Data are presented as the mean ± S.D. One-way ANOVA followed by Tukey’s post hoc test or Kruskal–Wallis test followed by Dunn’s post hoc test was used (n = 5 rats per group), * *p* < 0.05, ** *p* < 0.01, **** *p* < 0.0001. Scale bars: 50 μm.

**Figure 4 brainsci-13-01442-f004:**
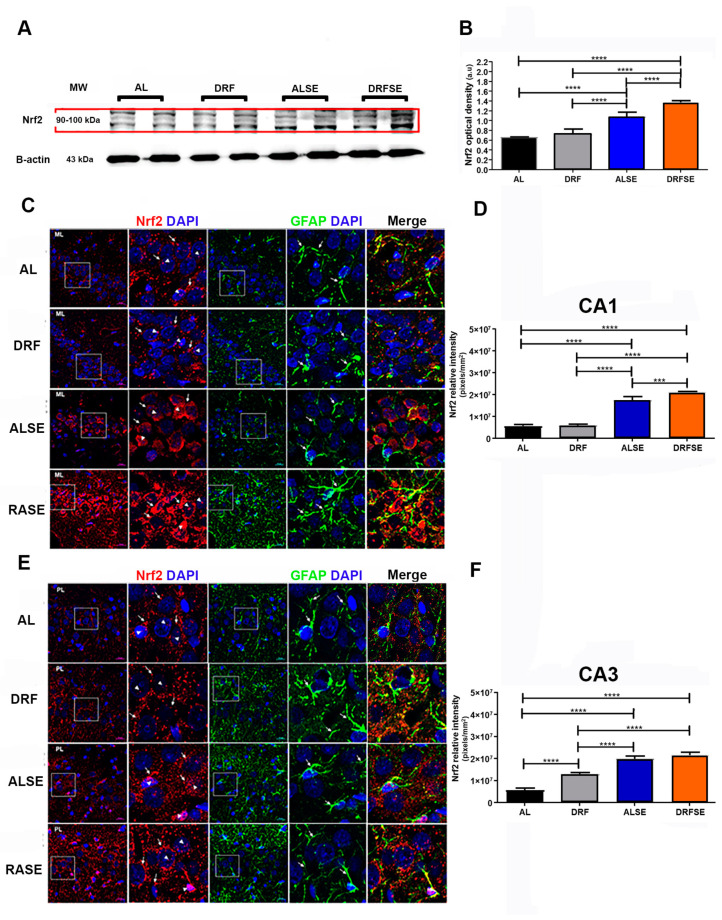
Effect of daytime-restricted feeding on the Nrf2 protein content and the Nrf2 and GFAP fluorescence in the CA1 and CA3 subfields of the dorsal hippocampus after SE induction. Representative immunoblots of Nrf2 in the AL, DRF, ALSE, and DRFSE experimental groups in the hippocampal homogenates (**A**). Quantification of the optical density of Nrf2 protein contents in hippocampal homogenates (**B**) Representative images of double immunofluorescence for Nrf2 (red) and GFAP (green) and counterstained with Hoechst (blue) in all experimental groups in the CA1 and CA3 subfields (**C**,**E**). Quantification of the relative intensity of Nrf2 in the CA1 subfield of all groups (**D**). Quantification of Nrf2 relative intensity in the CA3 subfield (**F**). Data are presented as the mean ± S.D, (n = 5 rats per group). Arrows show the cytoplasmic and head arrows show nuclear the distribution of Nrf2, respectively. One-way ANOVA followed by Tukey’s multiple comparison test, *** *p* < 0.001, **** *p* < 0.0001. Scale bars: 20 μm; ML: molecular layer.

**Figure 5 brainsci-13-01442-f005:**
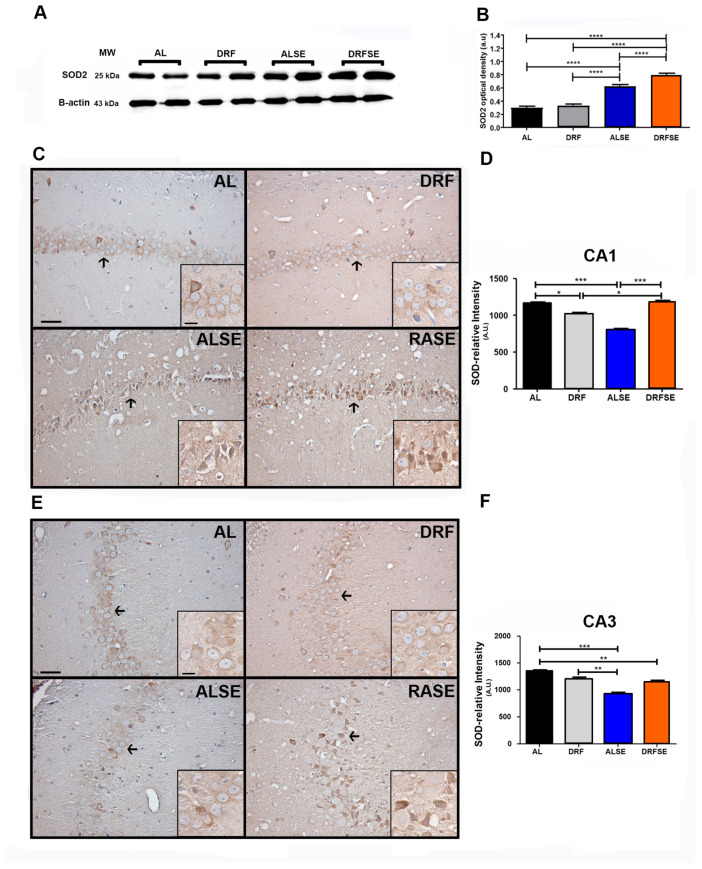
Effect of daytime-restricted feeding on the SOD2 protein content and the pyramidal distribution of SOD2 in the CA1 and CA3 subfields of the hippocampus after SE induction. Representative immunoblots of the SOD2 enzyme in the AL, DRF, ALSE and DRFSE experimental groups in the hippocampal homogenates (**A**). Quantification of the optical density of SOD2 protein contents in hippocampal homogenates (**B**). Representative photomicrograph of SOD2 immunoreactivity in the pyramidal cell layer of the CA1 subfield in all experimental groups (the arrows show the higher magnification image) (**C**) and the CA3 subfield (**E**). Quantification of SOD2 relative intensity in the CA1 (**D**) and CA3 subfields (**F**). Data are presented as the mean ± S.D, (n = 5 rats per group). One-way ANOVA followed by Tukey’s multiple comparison test, * *p* < 0.05 ** *p* < 0.01, *** *p* < 0.001 **** *p* < 0.0001. Scale bars: 20 and 50 μm.

## Data Availability

The data generated in the present study may be requested from the corresponding author.

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
