# Peer review of "Daytime-Restricted Feeding Ameliorates Oxidative Stress by Increasing NRF2 Transcriptional Factor in the Rat Hippocampus in the Pilocarpine-Induced Acute Seizure Model"

_brainsci, 2023, doi:10.3390/brainsci13101442_

Round 1

Reviewer 1 Report

The authors examined the possible protective effects of daily feeding restriction on epilepsy-induced oxidative stress. Using biochemistry, immunohistochemistry assays, they found daily restricted feeding could reduce oxidative stress in the hippocampal CA1 and CA3 regions, and this came along with the enhancing effect on Nrf2 antioxidant activity; unexpectedly they found a region-dependent change of SOD2 in epilepsy CA1 and CA3, while it was still increased by daily restricted food. The research and findings are intriguing and provide important insights and clues of how lifestyle and behavior change such as daily restricted feeding may benefit the epilepsy treatment.

Line 61, “several antioxidant enzymes” is not precise, hundreds of genes are direct targets of Nrf2.

Section 2.1, An institute approved animal care and use guideline for animal procedure is required. If N=5 animals per group, why used 80 rats? Describe briefly the study design.  

Line 89-96, the description is unclear and confusing for which group received the treatments/injections. A figure showing the treatment and injection of chemicals for each figure will be very helpful.

 Line 100, explain briefly why only used animals at stage 4, not others.

Line 221, DHE is a general assay marker for ROS but cannot specify whether the substrate is superoxide, hydrogen peroxide, or hydroxyl radical.

Figure 4, 1) the font of labels for the column panels are too small to be seen; 2) change of Nuclear Nrf2 is not shown but discussed in results and discussion.

Figure 5, use arrow to show SOD staining, CA1 and CA3 layers, pyramidal cells, and the magnification.

Line 64 and 294, citation for SOD2 is downstream target of Nrf2 is inappropriate.  

Some minor grammer errors.

Author Response

  1. Thank you for your comments. We corrected the sentence in the manuscript. We also wrote a sentence explaining that the Ethical committee of our institution approved our study.
  2. We used eighty animals to perform all the experiments. Twenty animals were used to perform malondialdehyde quantification, twenty animals were used to perform dihydroethidium quantification (we needed fresh tissue to incubate the DHE solution), twenty animals for the immunohistochemistry and immunofluorescence, and twenty animals for the western blot technique.
  3. We corrected the mistake in the description of the DRF schedule and made a figure explaining the experimental design.
  4. We only used animals that reached stages 4 to 5 because in these stages, limbic seizures are characteristics of status epilepticus (Turski, et al., 1983, doi: 10.1016/0166-4328(83)90136-5). Furthermore, most articles that study pathological, physiological, or biochemical changes in status epilepticus (SE), use animals that reach stage 4 or 5 because they are more likely to have recurrent seizures characteristic of SE.
  5. We agree with you. Dihydroethidium has been described in the literature as a superoxide ion indicator because can permeate cell membranes and be oxidized by cellular superoxide ions. However, it produced two red fluorescent products, the ethidium (E+) produced by a non-specific redox reaction (it can be produced by hydrogen peroxide or hydroxyl radical) and 2-hydroxy ethidium (2-OH-E+), a specific adduct produced by the reaction of DHE and superoxide ion. We cannot discriminate both products in epifluorescent microscope (Texas red filter) because the fluorescent spectrum of 2-OH-E+(Ex 500–530 nm/Em 590–620 nm) and E+ (Ex 520 nm/Em 610 nm) is very similar (Wang and Zou, 2018, doi:10.1007/978-1-4939-7598-3_32). According to the above information, part of the fluorescence emitted by DHE may be due to the reaction of DHE with the superoxide ion, but another part may be produced by other ROS.
  6. We corrected the size of the font in Figure 4.
  7. We discuss the nuclear distribution in the results and discussion section because, in the immunofluorescence figure, some pyramidal neurons are immunostained with Nrf2 antibody.
  8. We corrected the figure and added a higher magnification image.
  9. We corrected the citation in the manuscript.

Reviewer 2 Report

The manuscript is generally well-written, however, there are some points need to be clear:

1- How long was Daytime-restricted feeding? 

2- I suggest the authors add a timeline for experimental design

3- when was racine test performed? how long were the animals observed?

4- Have the authors detected blood or brain glucose levels?

minor errors

Author Response

  1. Thank you for your comments. The day-time restricted feeding lasted 20 days. We followed your recommendation to write a sentence in the manuscript specifying the diet schedule.
  2. We agree with your suggestion and made a figure explaining the experimental design.
  3. We monitor the animals for two hours. As described in the methodology, we only used animals that reached stages 4 to 5 because, in these stages, limbic seizures are characteristics of status epilepticus (Turski, et al., 1983, doi: 10.1016/0166-4328(83)90136-5). Furthermore, most articles that study pathological, electrophysiological, or biochemical changes in status epilepticus (SE), use animals that reach stage 4 or 5 because they are more likely to have recurrent seizures characteristic of SE.
  4. Yes, we have quantified glucose levels in experimental groups. In previous work, we found that DRF did not change glucose levels in the blood, however, we did find that the restrictive diet increased the ketone body beta-hydroxybutyrate levels in the blood (Landgrave-Gomez et al. 2016, doi: 10.1016/0166-4328(83)90136-5).